# Medicinal Values and Potential Risks Evaluation of *Ginkgo biloba* Leaf Extract (GBE) Drinks Made from the Leaves in Autumn as Dietary Supplements

**DOI:** 10.3390/molecules27217479

**Published:** 2022-11-02

**Authors:** Xiaojia Su, Ruirui Shi, Haiyan Hu, Linfeng Hu, Qichao Wei, Yuanyuan Guan, Jingling Chang, Chengwei Li

**Affiliations:** 1College of Life Science and Technology, Henan Institute of Science and Technology, Xinxiang 453000, China; 2Henan Engineering Research Center of Crop Genome Editing, Henan International Joint Laboratory of Plant Genetic Improvement and Soil Remediation, Xinxiang 453000, China; 3College of Chemistry and Chemical Engineering, Henan Institute of Science and Technology, Xinxiang 453000, China; 4College of Biological Engineering, Henan University of Technology, Zhengzhou 450001, China

**Keywords:** GBE drinks, dietary supplements, quality, pharmacodynamic ingredients, ginkgolic acids, acceptable daily intake

## Abstract

Ginkgo tea and ginkgo wine are two familiar *Ginkgo biloba* leaf extract (GBE) drinks in the form of dietary supplements (DS) used for healthcare in east Asia. Nevertheless, a comprehensive evaluation of their safety and efficacy is still lacking. In this study, GBE drinks were prepared from naturally newly senescent yellow leaves (YL) and green leaves (GL) in autumn. Their total flavonoids, antioxidant capacity and prescribed ingredients were investigated. In brief, the proportions of total flavonoids, total flavonol glycosides (TFs), total terpene trilactones (TTLs) and ginkgolic acids in the GBE drinks all did not meet the standards of worldwide pharmacopoeias. Specifically, the levels of TFs in the ginkgo tea prepared from YL were significantly higher than that prepared from GL. Further analyses revealed a substandard ratio of isorhamnetin/quercetin and an accumulation of leaf-age-related compounds, which were both unqualified. The proportions of specific TTLs varied between the ginkgo tea and ginkgo wine, although no significant differences were detected in terms of the total levels of TTLs. Noticeably, numerous biflavones and thousands of times over the limiting concentration of ginkgolic acids, including newly identified types, were only detected in ginkgo wine. Finally, the use of the GBE drinks as DSs was comprehensively evaluated according to the acceptable daily intake. This study showed the limited healthcare effects of GBE drinks despite their powerful antioxidant capacity.

## 1. Introduction

*Ginkgo biloba* (L.), the only surviving species of the Ginkgoaceae family, is widespread almost worldwide and is well-known for its *G. biloba* leaf extract (GBE). Owing to the unique pharmaceutical functions of GBE in the treatment of many degenerative diseases (e.g., Alzheimer’s disease), GBEs are one of the currently best-selling commercialized herbal products [1,2,3]. Dietary supplements (DS) are usually used for healthcare in traditional medical therapies [4]. Elders in China are always concerned with the medicinal values and potential risks of homemade ginkgo leaf products (ginkgo tea and ginkgo wine) as GBE DSs when staring at the ginkgo trees surrounding them [5,6]. Ginkgo leaf can generally be used to make tea, herbal extracts and tinctures [7]. Tea, made from different natural plants, has been traditionally used as a medication based on personal experience in China and Japan [8]. Commercialized ginkgo tea, made from ginkgo leaves, is processed through the traditional handicraft techniques of de-enzyming, rolling and drying [9]. During the process, the content of flavonoid glycosides decreases sharply [10,11]. Thus, it is reasonable that ginkgo leaves are collected and immersed in hot water as homemade ginkgo tea without further processing in China considering the active ingredients content [12].

Commercialized ginkgo wine is usually made from ginkgo nuts mixed with ginkgo leaves. They are initially milled and are then mixed with other raw materials, and finally they are manufactured into ginkgo wine following the typical Chinese Baijiu (distilled spirits of China) procedures (fermentation, distillation and blending) [13]. It is generally considered safe considering the higher antioxidant capacity and the limited contents of hydrocyanic and ginkgolic acids [14,15]. Ginkgo leaves are the GBE raw materials and contain abundant pharmaceutical ingredients. Thus, it is also common that homemade ginkgo wine is manufactured by directly immersing ginkgo leaves in Chinese Baijiu, just like the method of processing most traditional critical herbal products (e.g., ginseng wine) in east Asia [12,16,17].

Researchers have controversial viewpoints on the safety of homemade ginkgo tea and ginkgo wine as daily DS drinks. Wang et al. [12] suggested that ginkgo tea is reasonable, but ginkgo wine should be prohibited because of its extremely high concentrations of ginkgolic acids. However, an accurate quantitative assessment and comprehensive analysis of each active ingredient needs to be conducted. Other researchers have called on the dangers of ginkgo tea due to its ginkgolic acid content and also potential herb–drug interactions [5,6,18]. In addition, the safety and efficacy of ginkgo wine made totally from ginkgo leaves have not been thoroughly investigated. However, the network has been flooded with information about its healthcare effects on elders, especially on patients suffering from cardiovascular and cerebrovascular diseases (http://m.zhongyoo.com/yaojiu/4842.html, accessed on 30 October 2022).

Medical research has revealed that total flavonol glycosides (TFs) (Figure 1a) and terpene trilactones (TTLs) (Figure 1b) are the two main pharmacoactive ingredients in GBE [19]. Meanwhile, ginkgolic acids are considered the primary injurious ingredient of GBE products. These compounds belong to the plant’s secondary metabolism and are easily affected by the leaf development stage, sampling time and surrounding environments [20,21,22].

Typically, the quality of GBE is also significantly affected by the leaf conditions and extraction solvents used [12,20]. Usually, the leaves in spring or summer are collected for commercialized ginkgo tea. However, green leaves in autumn are usually stipulated for various ginkgo-leaf-derived medicines or products according to the Chinese pharmacopoeia [19,23]. Horbowiczt et al. [22] showed that the major TTL (ginkgolides B and C) levels were significantly higher in naturally senescent yellow leaves than green leaves on 9 September, which indicates the importance of the senescence process on TTL accumulation. Thus, the usability of leaves (especially the naturally senescent yellow leaves) in autumn for processing GBE drinks needs to be investigated thoroughly.

In addition, the free radical scavenging ability, to some extent, reflects the potential healthcare effects of the tested compounds. Wang et al. [15] evaluated the antioxidant capacity of commercialized ginkgo wine by DPPH and ABTS assays. However, related studies about homemade GBE drinks are rare.

To the best of our knowledge, few studies have been put forward on the safety and efficacy evaluation of homemade GBE drinks made from natural ginkgo leaves in autumn, although it is a widespread concern in east Asia. Hence, the main goal of this study was to comprehensively evaluate them by assessing the representative ingredients through HPLC-MS/MS and UHPLC-QTOF-MS and also by assessing their antioxidant capacity. Then, by integrating and comparing the accurate data with industrial criteria, the healthcare effects of the use of GBE drinks as DSs was assessed. This research will be helpful in clarifying the confusing conjecture on the safety and efficacy of GBE drinks and also in improving our knowledge of the processing and utilization of ginkgo leaves in autumn.

## 2. Results

### 2.1. Total Flavonoids Content and DPPH and ABTS Free Radical Scavenging Activity of GBE Drinks

Green ginkgo leaves (GL) in late October and naturally newly senescent yellow leaves (YL) in early November were collected for further preparation (Figure 2a). Homemade ginkgo tea and ginkgo wine were prepared as GBE drinks according to the description in Section 4.2. There were apparent differences in the appearances of the GBE drinks. The ginkgo tea appeared to have varied cloudy yellow colors. In contrast, the ginkgo wine presented varied clear yellow hues (Figure 2b). Additionally, ultrasound-assisted extraction could make darker shades of tea or wine, which indicated more effective extractions (Figure 2b).

The ginkgo tea and ginkgo wine from ginkgo leaf powder were investigated for total flavonoids. Comparable levels (*p* = 0.46) of total flavonoids in the ginkgo tea from YL (6.63 ± 0.63 mg/g·DW) were detected compared with GL (6.27 ± 0.44 mg/g·DW) without ultrasound assistance (Figure 2c). Meanwhile, significant increases were both found in the ginkgo tea (*p* = 0.008) and ginkgo wine (*p* = 0.028) produced under ultrasound-assisted extractions, especially using the YL (Figure 2c). This might be explained by the different surface morphologies and wettability of the GL and YL [24], through which the liquids could saturate the dried GL easier even without the ultrasound assistance. These demonstrated the positive effects of the ultrasound treatment on flavonoid extraction, especially with YL. Additionally, the levels of total flavonoids in the GBE drinks from YL were significantly higher (*p* = 0.047 for ginkgo tea and *p* = 0.026 for ginkgo wine) than in the drinks from GL under ultrasound-assisted extraction (Figure 2c).

Polyphenols contribute to the main antioxidant properties of traditional wine and tea [15,25]. Among them, total flavonoids play essential roles. DPPH and ABTS assays were performed to determine the antioxidant capacity of the GBE drinks. The antioxidant capacities of the GBE drinks are listed in Table 1, and water and Chinese Baijiu (Luzhou Laojiao) were used as controls. The free radical scavenging capacities of the ginkgo tea from YL determined by DPPH and ABTS were 133.59 and 578.95 μg Trolox/mL, respectively, which were significantly higher (*p* < 0.01) than those of the drinks from GL under ultrasound-assisted extraction (Table 1). There was also the same tendency in the ginkgo wine (Table 1). The average levels of the free radical scavenging capacities of the GBE drinks were significantly higher (*p* < 0.01) than those of the Chinese Baijiu (Luzhou Laojiao) and pure water (Table 1), which was consistent with the total flavonoid contents of the GBE drinks (Figure 2). The free radical scavenging capacities of the tested compounds, to some extent, reflected their potential health care effects. The results in the present study indicated that GBE drinks might have better health care effects than the tested Chinese Baijiu (Luzhou Laojiao), and the GBE drinks from YL had advantages over the drinks from GL.

### 2.2. Total Flavonols (TFs) in GBE Drinks

Total flavonol aglycones (quercetin, kaempferol and isorhamnetin) were examined and quantified by acidification as described before [26]. The method was tested, through which quercetin-3-*O*-glycoside (upper panel) was hydrolyzed by 6 N HCl and then turned into quercetin (Appendix A). The levels of total flavonols in the ginkgo tea from YL (4.54 ± 0.29 mg/g·DW) were significantly higher (*p* = 0.02) than the tea from GL (3.61 ± 0.16 mg/g·DW), and the ultrasound did not have an observable effect (Figure 3a). The total flavonol level in the ginkgo wine from YL (5.76 ± 0.21 mg/g·DW) was 26.9% higher (*p* = 0.004) than that in the ginkgo tea under ultrasound-assisted extraction. Meanwhile, the level of total flavonols in the ginkgo wine from GL was 45.3% higher (*p* = 0.002) than that in the ginkgo tea (Figure 3a), although the total flavonoid levels of them both showed no significant differences (Figure 2c). This suggested that the Chinese Baijiu was more conducive to leaching out the total flavonols than water under ultrasound-assisted extraction. This was similar to previous research, in which the levels of total flavonols in 40–70% ethanol infusions were significantly higher than in pure water [12].

The proportions of the three main flavonol aglycones (quercetin, kaempferol and isorhamnetin) are also explicitly stipulated in the Chinese pharmacopoeia [19]. The contents of the three flavonol aglycones are listed in Table 2. The proportions of quercetin/kaempferol in the ginkgo tea (1.04–1.06 for YL; 1.01–1.03 for GL) and wine (1.01–1.02 for YL; 0.94–0.95 for GL) were both in keeping with the criterion (0.8–1.2) of the Chinese pharmacopoeia (Figure 3a) [19]. In contrast, the proportions of isorhamnetin/quercetin were only around 0.048–0.062 in both the GBE drinks, which were far below the criterion of 0.15 (Figure 3a). This might be attributed to the susceptibility and instability of isorhamnetin glycosides [27].

Furthermore, the HPLC fingerprint of total flavonol glycosides (TFs) in GBE is also stipulated as another quality criterion in the Chinese pharmacopoeia [19]. The TF profiles analysis showed no substantial differences, but the distinct advantages of biflavones (peak 2–5, bilobetin, ginkgetin, isoginkgetin and amentoflavone) were detected in the ginkgo wine compared with the ginkgo tea (Figure 3b). Biflavones might endow the ginkgo wine with great healthcare values due to their potential pharmaceutical values [28,29]. In addition, the content of a flavonoid-like compound with a typical flavonol ultraviolet absorption spectrum of ~250 nm and ~360 nm (marked as peak 1) at 9.94 min significantly increased by about 30% in the YL compared to GL both in the ginkgo tea and wine, but it was not detected in the standard EGb761 (Figure 3b and Figure 4a). To further analyze the accumulation profiles of this compound, we determined the flavonol glycoside levels of ginkgo leaves from April to November. The results showed that it was a leaf-age-related compound, which rose from August and then increased rapidly during the maturation and senescence of the leaf (Figure 4b). Similarly, some researchers have shown that specific flavonoids (e.g., quercetin-4-glucoside and rutin) accumulate at a higher level in naturally senescent yellow ginkgo leaves than in green leaves [22]. All these suggested that ginkgo leaves in autumn can accumulate some specific flavonols that might be absent in fresh leaves. However, their medicinal values and edible safety need more investigation.

### 2.3. Terpene Trilactones (TTLs) in GBE Drinks

Ginkgolides (A, B, C, J, N, K and M) and bilobalide are the terpene trilactones that have been mainly detected in *G. biloba* up to now [2]. The contents in native ginkgo leaves are less than 0.06%, among which ginkgolide A, B and C are the quantitatively dominant compounds [2,22]. Ginkgolide A (7.49 min, *m*/*z* = 407.4), B (8.0 min, *m*/*z* = 423.3), C (5.9 min, *m*/*z* = 439.3), J (5.76 min, *m*/*z* = 423.1) and bilobalide (5.8 min, *m*/*z* = 325.3) were detected in the ginkgo tea and ginkgo wine, and this was calculated by using UHPLC-QTOF-MS (Appendix A). There were no significant differences (*p* > 0.05) in the TTL contents between the ginkgo tea (252.89 ± 6.24 μg/g·DW) and ginkgo wine (248.2 ± 4.93 μg/g·DW) (Figure 5). Among the TTLs detected, ginkgolide A, B and J were the quantitatively dominant compounds followed by ginkgolide C and bilobalide (Figure 5). Additionally, the ultrasound-assisted extraction significantly increased the levels of TTLs both in the ginkgo tea and wine (*p* < 0.05) (Figure 5).

Although no big differences were detected in the total TTL contents of the ginkgo tea and ginkgo wine, specific categories varied. The levels of ginkgolide A (84.04 ± 1.87 μg/g·DW) and ginkgolide J (72.12 ± 5.02 μg/g·DW) in the ginkgo wine were significantly higher (*p* = 0.002 and *p* = 0.03) than in the ginkgo tea (ginkgolide A, 79.39 ± 4.28 μg/g·DW; ginkgolide J, 68.27 ± 3.13 μg/g·DW). In contrast, the levels of ginkgolide B (47.04 ± 1.05 μg/g·DW), ginkgolide C (26.62 ± 4.26 μg/g·DW) and bilobalide (18.45 ± 0.79 μg/g·DW) in the ginkgo wine were significantly lower (*p* < 0.01 or 0.05) than in the ginkgo tea (Figure 5, Table 3). No significant differences were detected between YL and GL, except for a significantly higher level (*p* = 0.02) in the ginkgo tea from YL without ultrasound assistance, which was, to some extent, consistent with former research [22] (Figure 5, Table 3).

### 2.4. Ginkgolic Acids in GBE Drinks

According to the Chinese pharmacopoeia, the total ginkgolic acids in GBE are explicitly limited to a limiting concentration of 5 mg/Kg considering their negative properties [2,19,30]. The total ginkgolic acids in the ginkgo tea and ginkgo wine were examined through authentic substrates (total ginkgolic acids A and B, including C13:0, C15:0, C15:1, C17:1 and C17:2) (Figure 6a, left) by HPLC-MS/MS (Appendix A). As many as ten peaks were detected in the ginkgo wine, and five were affirmed as C13:0, C15:1, C17:2, C15:0 and C17:1 in sequence. Among them, the ginkgolic acids of C15:1 and C17:1 were the quantitatively dominant compounds, followed by C13:0, C17:2 and C15:0 (Figure 5a, right, upper panel; Appendix A). Only traceable levels of C15:1 and C17:1 were detected in the ginkgo tea even under the ultrasound-assisted extraction, both of which were familiar remnants from other ginkgo leaf products [31] (Figure 6a, right, lower panel).

Further quantitative analysis showed that the concentration of the total ginkgolic acids in the ginkgo wine was as high as 6.77 ± 0.25 mg/g·DW, which was significantly higher than that without ultrasound treatment (1.69 ± 0.35 mg/g·DW) (*p* < 0.01) (Figure 6b). In addition, the content of ginkgolic acids in the YL (1.69 ± 0.35 mg/g·DW) was at a relatively higher level than the GL (1.08 ± 0.29 mg/g·DW) even without ultrasound assistance. These suggested that the ultrasound severely increased the concentrations of the total ginkgolic acids in the ginkgo wine and YL were inclined to accumulate more ginkgolic acids. The content of total ginkgolic acids in the ginkgo wine was equal to 1080–6770 mg/kg·DW. It was thousands of times over the limiting level (5 mg/kg·DW) stated in the Chinese pharmacopoeia.

The ginkgolic acids profiles showed that 11 kinds of ginkgolic acids (namely, C13:0, C14:0, C15:0, C15:1-8Z, C15:1-10Z, C17:0, C17:1-8Z, C17:1-10Z, C17:1-12Z, C17:2-8Z-11Z and C17:3) have been identified in ginkgo up to now [2,32]. Except for the five identified ginkgolic acids from the authentic substrates, five new peaks presenting similar ultraviolet absorption spectra (~250 nm and ~310 nm) with the characterized ginkgolic acids in the ginkgo wine being discovered and numbered 1 to 5 (Figure 6a and Appendix A). Furthermore, the relative molecular masses (RMM) at the ESI^-^ mode of the five compounds (1–5) were 361.21, 389.20, 387.14, 317.05 and 369.12, respectively (Table 4, Appendix A). According to the structures and naming principles of ginkgolic acids, we deduced the five probably new ginkgolic acids as having alkyl substituents of C16:0, C18:0, C18:1, C13:1 and C17:3 (Table 4). For the identified ginkgolic acids, a single double bond was usually formed at the C-8′ position and two double bonds were generally formed at the C-8′ and C-11′ positions, respectively [2]. Additionally, Quenon et al. [33] identified a ginkgolic acid compound with a C17:3 aliphatic chain from *Syzygium malaccense* (L.), as did Merr. and L.M. Perry with an *m*/*z* value of 369.2, in which three double bonds were formed at the C-8′, 11′ and 14′ positions through confirmation using 2D NMR (nuclear magnetic resonance). The supposed structures of the five newly identified ginkgolic acids are shown in Appendix A [2,32,33].

## 3. Discussion

As an alternative to traditional medical therapies, dietary supplements (DS) are widely used worldwide [4,34]. Tea and functional wines, as daily health drinks in China, have been traditionally used as medications based on personal experience for thousands of years [8]. Meanwhile, considering the wide distribution of *G. biloba* in China, homemade GBE drinks have been popular among elders [5,6,12]. However, comprehensive evaluations of their efficacy and safety are still lacking. Moreover, controversial and even misleading viewpoints exist extensively in China. Therefore, the main goals of the present study were to investigate the levels of the major active ingredients (TFs and TTLs) and toxic components (ginkgolic acids) in homemade GBE drinks, which are the essential quality control standards of GBE products worldwide [19,30]. Then, the safety and efficacy of the homemade GBE drinks were evaluated.

In the present study, homemade ginkgo tea and ginkgo wine from China were prepared according to experience described before [5,6,12]. Green leaves in autumn are recommended for collection for industrial processing [19,35]. Naturally senescent yellow leaves are considered to contain a higher level of some specific compounds (e.g., terpene trilactones) [22]. Meanwhile, it is common for elders to collect ginkgo leaves in late autumn for personal use in China. Additionally, considering the experience used among elders and the extraction efficacy in the food industry, GL and YL powder in autumn with ultrasound treatment were selected to prepare the GBE drinks [25,36]. The prepared GBE infusion presented a varied yellow color, which was consistent with that of commercial ginkgo tea [9].

The contents of total flavonoids in drinks and plant extracts are believed to be associated with their health-promoting properties [37,38]. Although the levels of the total flavonoids are not mentioned in pharmacopoeias of Chp2020, EP8.0 and pharmacopeia USP36, they are always required for traditional Chinese medicine production and the management standards of GBE50 [19,39,40]. In our study, the levels of total flavonoids in the GBE drinks were about 4.83–8.58 mg/g·DW, which were equal to 12.08–21.45% under a solid-to-liquid ratio of 1:40. The values were significantly lower than the criterion (44.0–55.0%) of GBE50 [40]. This suggested that the homemade GBE drinks from autumn in this study were substandard GBE products, considering the levels of the total flavonoids. Additionally, the levels of total flavonoids from YL were significantly higher than that from GL, which was consistent with the tendency of antioxidant capacities (Figure 2c). Meanwhile, the antioxidant capacities of the ginkgo tea and ginkgo wine were higher than those of the typical Chinese Baijiu tested in this study, which was in line with former research [15]. This suggested that homemade GBE drinks obtain their potential healthcare effects by the process of antioxygenation, but the content of total flavonoids was significantly lower than the criterion of GBE50 in this study.

TFs and TTLs both belong to the plant’s secondary metabolism and are easily affected by the genetic background, leaf development stage, sampling time and surrounding environments [20,21,22]. Researchers have shown that TFs and TTLs in ginkgo leaves vary over one year. The levels of TTLs increase gradually from May, then reach a peak in August, and subsequently decrease in late November, during which a relatively high level is observed in autumn (October/November) compared to spring (May) [41]. In the present study, the GBE drinks from YL had comparative advantages over the drinks from GL in terms of the content of active ingredients (Figure 3a and Figure 5). However, the levels of TFs (4.54 mg/g·DW) and TTLs (0.256 mg/g·DW) in the ginkgo tea from autumn leaves were substantially lower than that derived from fresh ginkgo leaf in May, the TFs and TTLs content of which was about 9.2 mg/g·DW and 1.21 mg/g·DW, respectively [9]. The active ingredient proportions in the GBE drinks were about 8.55–14.4% for TFs and 0.61–0.65% for TTLs, which were all far below the GBE standards (≥24% for TFs and ≥6% for TTLs) of pharmacopoeias worldwide, especially the levels of total TTLs [19,39]. In our study, the proportion of BB/(GA+GB+GC) was around 0.11–0.14 (Figure 5), which was out of the criterion of 0.42–2.07 as stated in USP41 [39]. Lu et al. [42] showed that the proportions of bilobalide (BB)/ginkgolides (GA+GB+GC) in eleven commercial ginkgo leaf preparations ranged from 0.3–0.8, and the proportion of 0.5 produced the maximum antithrombotic effects. All these suggested the discounted healthcare effects of the present GBE drinks. Additionally, former research has shown that the senescence process might contribute to TTL accumulation [22]. However, in the present study, the levels of TTLs in the GBE drinks from GL and YL were similar, which might be ascribed to the sampling period of the control samples. Considering the comparative advantages of YL in terms of the levels of active ingredients over GL, YL in autumn is worth exploring rather than them being wasted [30].

For ginkgolic acids, a mixture containing 13–19 straight alkyl side chain carbon atoms with 0 to 3 side chain double bonds at site 6 of ortho hydroxy benzonic acid is considered the primary injurious part of ginkgo leaf standard extracts [2,19]. In this study, the concentration of ginkgolic acid in the ginkgo wine was as high as 6.77 ± 0.25 mg/g·DW, which was thousands of times over the limiting concentration of 5 mg/kg for all alkylphenols in the European, U.S. and Chinese pharmacopoeias [19,30]. Therefore, homemade ginkgo wine in China should be strictly prohibited due to its unsafety. Up to now, as many as eleven different ginkgolic acids have been identified [2,32]. In our study, another five kinds of ginkgolic acids with specific RMMs were detected, with one of them being identified as C17:3 according to related studies [32,33]. The other four were identified as ginkgolic acids with alkyl side chains at C16:0, C18:0, C18:1 and C13:1 based on the RMMs, the characteristics of their molecular structure and the naming rules of the known ones (Table 4). These might be helpful for studies on ginkgolic acids. The ADI (acceptable daily intake) of TFs and TTLs for disease treatment are stipulated as 57.6 mg and 14.4 mg according to the instructions of ginkgo leaf prescription drugs, respectively (Yinxingye Pian with an SFDA approval number Z20053912). In the present study, 56.35 g YL powder was infused with boiled water at a solid-to-liquid ratio of 1:40, and about 2254 mL of the first tea infusion contained 254.2 mg of TFs and 14.4 mg of TTLs, which just precisely met the GBE ADI, especially for the TTLs (Figure 7), whereas the Chinese Dietary Guidelines (2022) recommend that the daily water intake for adults in China should be 1500 to 1700 milliliters. In addition, a new Lancet study showed that elder individuals (≥40 years) with moderate alcohol consumption (30–60 mL) can be healthy [43]. Correspondingly, only 8.7 mg of TFs, 0.38 mg of TTLs, and as high as 10.15 mg of ginkgolic acids were detected in 60 mL of the ginkgo wine, which possessed minimal pharmacodynamic substances but had huge potential risks according to the Chinese pharmacopoeia [19]. This suggested that it might be challenging for elders to obtain a total healthcare dosage through only ginkgo tea or ginkgo wine for disease treatment. Furthermore, the edible safety of the substandard ratio of isorhamnetin/quercetin and other leaf-age-related compounds in ginkgo tea will need more confirmation before clinical application (Figure 4). Importantly, it is generally alarming that *G. biloba*’s concurrent use with some prescription drugs by elders might have some potential risks such as bleeding [44].

Finally, ginkgo drinks are welcomed mainly for their possible healthcare functions. However, ginkgo tea has not been widely marketized, particularly because of its bad palatability and vague functional statement [8]. Thus, up to now, there has been no national or industrial standard for this traditional Chinese medicinal material as a functional drink. In addition, several kinds of ginkgo tea are sold as non-processed agricultural products without functional interpretation by Alibaba (Hangzhou, China). Therefore, ginkgo leaves are used mostly according to the purchasers’ experience. Ginkgo wine made from the ginkgo plant is a liqueur (called Lu Jiu in China). Liqueurs are extractions of MF (medicine and food)—Chinese medicinal herbs with Chinese rice wine or Baijiu. Original styles of the wine have changed, and more attention has been given to the palatability, nutritional and tonic functions of ginkgo liqueurs. Ginkgo nuts are usually the raw material used in ginkgo wine, and the level of TFs tested in the Chinese pharmacopoeia is only stipulated to be at least 0.3 mg/L (Q/LYGZT 0001S—2021), which is far lower than the ginkgo wine we prepared from ginkgo leaves in this study (~145 mg/L). However, considering the ginkgolic acids in the ginkgo wine produced from leaves, the liqueur from ginkgo leaves has been less investigated. Fan and Kuang [45] tried to make it with materials of ginkgo leaves, the fruit of the Chinese wolfberry and jujube (m/m/m = 5:1:1), and the total flavonoid content of the mixed drinks was as high as 2460 mg/L. However, the potential side effects of the ginkgolic acids in the drinks were not considered. Despite the mixed raw material, they provided information on the more significant advantages of the TF level in ginkgo wine from ginkgo leaves compared to ginkgo nuts. To our interest, if the ginkgolic acids can be removed, ginkgo wine made totally from ginkgo leaves can be a potential product or raw material of GBEs due to their beautiful appearance, considerable number of TFs, TTLs and high level of biflavones. Inorganic ceramic membranes, one category of nanofiltration membranes, can eliminate the low thermal and chemical stability and low mechanical strength of organic membranes [46,47]. Ling et al. [48] discovered a ceramic membrane preparation method, through which 99.9% of ginkgolic acids can be removed from the ethanol extract of *G. biloba* leaves. This technology might be an excellent choice for eliminating the ginkgolic acids from homemade ginkgo wine for its future industrial production. 

## 4. Materials and Methods

### 4.1. Plant Materials and Reagents

The *Ginkgo biloba* leaves in this study were collected from a tree (diameter at breast height = 16 cm) growing in the Henan Institute of Science and Technology (35°16′44″ N; 113°56′11″ E) (Xinxiang, Henan Province, China) in 2021. The leaf petioles were removed, and only the leaf blades were cleaned and dried naturally at room temperature with a constant weight for further analysis.

All authentic substrates were purchased from Tauto Biotech Co., Ltd. (Shanghai, China). All solvents used for HPLC-MS/MS were chromatographically pure from Shield (Tianjin, China) except for the formic acid, which was analytical grade. The water used in this study was prepared through a Milli-Q water-refining system (Millipore, Billerica, MA, USA). Prescription medicine “Yinxingye Pian” with an SFDA approval number Z20053912, as a standard of EGb761, was produced by Jiangsu Runbang Pharmaceutical Co., Ltd. (Huai’an, China).

### 4.2. Preparation of the Ginkgo Tea and Ginkgo Wine

The ginkgo tea and ginkgo wine were prepared as follows. Part of the naturally dried ginkgo leaf blades were ground into a powder with a crusher and then boiled in ultrapure deionized water with a solid-to-liquid ratio of 1:40 for 4 h and then finally cooled naturally overnight. The ultrasonication was conducted twice for 30 min each by a KQ3200 supersonic cleaner (150 W, 40 KHz) made by Kunshan Ultrasonic Instruments Co., LTD. (Kunshan, China). Then, the ginkgo tea was the supernatant obtained through three repetitions of centrifugation at 12,000 rpm for 10 min.

Correspondingly, the ginkgo wine was prepared similarly to the ginkgo tea with a replacement of boiled water with Chinese Baijiu (Luzhou Laojiao with 52% alcohol). All the prepared products were filtered through 0.22 μm membranes. All the extractions were implemented with three technical replications.

### 4.3. Total Flavonoids Determination

Total flavonoids were quantitatively determined by chemical colorimetry methods with 5%NaNO_2_–10%AlCl_3_–1 M NaOH [49]. Rutin was used as a positive control for the standard curve, and a microplate reader (Thermo Scientific, Waltham, MA, USA) was used to collect the absorbance at 510 nm. Total flavonoids contents were calculated according to the standard curve (y = 0.001x + 0.0097, R^2^ = 0.9981, where y and x represent the absorbance value and total flavonoid concentrations, respectively).

### 4.4. Free Radical Scavenging Capacity Assays

To evaluate the free radical scavenging capacities of the GBE drinks, DPPH (DPPH·) and ABTS (ABTS·) kits were purchased from Suzhou Grace Biotechnology Co., Ltd. (Suzhou, China). The infusions of the GBE drinks were diluted four times and then used for the determination according to the manufacturer’s instructions. The absorbances of the mixed solutions were recorded at 517 nm for the DPPH· and 734 nm for the ABTS· assays. The inhibition percentages of the DPPH·(ABTS·) rate were calculated as follows:Inhibition_sample_ (%) = [1 − (Abs_sample_ − Abs_control_)/Abs_blank_] × 100%
where Abs_control_ represents the absorbance of the control sample. In the same way, the Abs_sample_ is the absorbance of the sample [15].

The DPPH· scavenging capacities of the samples were calculated as follows:DPPH· scavenging capacity (μg Trolox/mL) = [(Inhibition_sample_ × 100 − 0.7084)/2.8486] × 4.

The ABTS· scavenging capacity of the samples were calculated as follows:ABTS· scavenging capacity (μg Trolox/mL) = [(Inhibition_sample_ × 100 + 1.4213)/0.5042] × 4.

The scavenging capacities of all samples were expressed as trolox equivalent antioxidant (TE). The standard solutions of trolox ranged from 0 to 25 mg/L for the DPPH· and 0 to 100 mg/L for the ABTS· assay. Two independent experiments were conducted, and each sample was analyzed in triplicate. The results represented one of the two independent experiments.

### 4.5. Flavonol Glycosides Profiles, Flavonol Aglycones and Biflavones Analysis

Flavonol glycosides profiles, flavonol aglycones and biflavones were analyzed by HPLC-MS/MS (Thermo Scientific, Waltham, MA, USA) equipped with a PDA detector, quaternary pump and an autosampler. Measurements were performed using an Inertsil ODS-3 C18 column (250 × 4.6 mm, 5 μm i.d.; GL Sciences Inc., Tokyo, Japan). The eluents used for separation were water containing 0.1% formic acid (solvent A) and 100% acetonitrile (solvent B). The elution gradient was as described before [49] with some changes: 0 min, 5% B, linear elution until 70% B at 30 min, 100% B at 35 min, then a return to 5% B for 1 min, and the initial gradient was maintained for 4 min. PDA data were recorded from 200 to 900 nm, and the 360 nm wavelength was set to a UV detector for testing the flavonols.

To analyze the total flavonol aglycones, the samples were treated with an equal volume of 6 N HCl at 70 °C for 40 min for hydrolysis, and twice the original sample volume of 100% methanol was added to stop the reaction [26]. Then, the mixture was injected for total flavonol aglycones analysis. Each aglycone was quantified and calculated with the specific standard curve plotted by HPLC with an authentic substrate (Kaempferol, y = 812885x – 50,099, R^2^ = 0.9972; Quercetin, y = 817,195x – 125,159, R^2^ = 0.9969; Isorhamnetin, y = 4 × 10^6^x, R^2^ = 0.9769; y represents the peak area). Total flavonol glycosides were calculated as per the description in the Chinese pharmacopoeia [19]. Compounds were further confirmed by authentic substrates and HPLC-MS/MS with *m*/*z* ranges from 80 to 1000 in a negative-ion (NI) mode. The MS detection conditions were as follows: desolvation temperature, 400 °C; desolvation gas (N2) rate, 800 L/h; cone gas flow, 50 L/h; and the cone voltage and capillary voltage were −60 V and 2 kV, respectively. Xcalibur software was used for system control and data processing. Total flavonol glycosides were calculated as per the description in Chinese pharmacopoeia [19].

### 4.6. Quantification of Terpene Trilactones

Terpene trilactones were quantified by an ultra-high-performance liquid chromatography ion-trap time-of-flight mass spectrometry (UHPLC-QTOF-MS) as described before [32] with some changes. For further analysis, authentic substrates (Ginkgolide A, B, C, J and bilobalide) were injected as positive controls. The mobile phase (delivered at 0.2 mL/min) comprised 0.1% formic acid water (A) and 100% acetonitrile (B). Gradient elution conditions were set as follows: 0–7 min, 5–30% B; 7–10 min, 30–50% B; and 10–12 min, 50–100% B. The total run time was 12 min. An Agilent 6540 Q-TOF mass spectrometer was operated for further determination. A negative mode electrospray ionization (ESI^-^) source was used with the follow conditions: gas temperature = 320 °C; sheath gas temperature = 350 °C; sheath gas flow = 11 L/min, nebulizer at 35 psi; and Vcap = 2500 V, nozzle voltage (Expt) = 1000 V. Qualitative Analysis B.07.00 was used for the data analysis.

### 4.7. Identification and Quantification of Ginkgolic Acids

Total ginkgolic acids were detected by HPLC-MS/MS as mentioned above with flavonol glycosides determination in Section 4.5 with an RD-C4 column (4.6 × 150 mm × 5 μm, Zhongpuhong, Zhongpu Science, Fuzhou, China) according to the methods described before [19,50] with some modifications. The mobile phase consisted of 0.1% formic acid water (A) and 100% acetonitrile (B). A linear gradient program was employed: 0~20 min, 70→90% B; 20~25 min, 90% B; 25~26 min, 90→70% B; and 26~30 min, 70% B. The flowrate was set to 1 mL/min and the column temperature was 30 °C. The UV detector was set to 200 to 900 nm, and chromatograms were obtained at 310 nm. The MS detection conditions were identical to that described in Section 4.5.

Ginkgolic acids were quantified and calculated with the specific standard curve plotted by HPLC-MS/MS with authentic substrates (C13:0, y = 144904x, R^2^ = 0.9935; C15:1, y = 707789x, R^2^ = 0.9969; C17:2, y = 23430x, R^2^ = 0.9949; C15:0, y = 62022x, R^2^ = 0.9985; C17:1, y = 648393x, R^2^ = 0.9992; y represents the peak area).

### 4.8. Statistical Analysis 

All the mentioned tests were performed in triplicate, and the data were expressed as mean values with standard deviation (S.D.) and Duncan in SPSS Statistics 17.0 with an alpha value at 0.05 and T.TEST in Excel were used to analyze the differences.

## 5. Conclusions

In our study, ginkgo tea and ginkgo wine made from GL in late October and YL in early November were evaluated for the well-known active ingredients of ginkgolic acids and antioxidant capacity. The results showed that 2254 mL of the first tea infusion derived from YL powder with a 1:40 solid-to-liquid ratio contained 254.2 mg of TFs and 14.4 mg of TTLs, which could meet the daily ADI of ginkgo leaf prescription drugs. However, the volume of tea obviously exceeded the 1700 mL daily water intake for adults in China recommended in the Chinese Dietary Guidelines (2022). Additionally, the edible safeties of unqualified isorhamnetin/quercetin and leaf-age-related compounds needed more confirmation. Compared with the ginkgo tea, an extreme excess of dozens of ginkgolic acids were probably the primary potential danger in the ginkgo wine according to industrial standards. Thus, the healthcare effects of the ginkgo tea and ginkgo wine were limited when we took the healthy daily water and ethanol intake into consideration. Noticeably, the GBE drinks from YL had comparative advantages over the drinks from GL in terms of the content of active ingredients. These suggested that YLs in autumn are worth exploring as a GBE material rather than them being wasted. Meanwhile, the considerable active ingredients of TFs, TTLs and biflavones might provide tremendous possibilities for the further exploitation of ginkgo leaf wine.

## Figures and Tables

**Figure 1 molecules-27-07479-f001:**
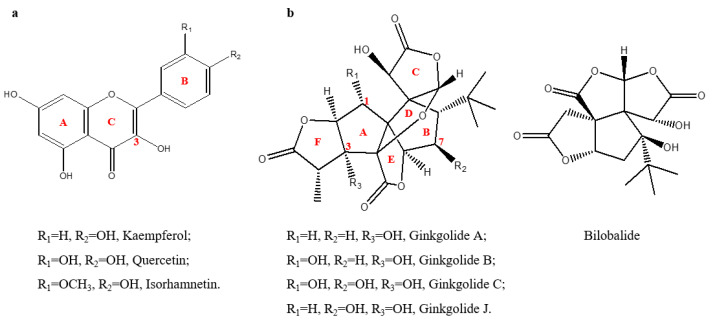
Structures of the major pharmacoactive ingredients in GBE. (**a**) Major flavonols (different glycosyls are usually linked at the 3-OH position). (**b**) Major terpene trilactones (ginkgolides and bilobalide). Structures were drawn using ChemDraw Ultra8.0 (CambridgeSoft, U.S.).

**Figure 2 molecules-27-07479-f002:**
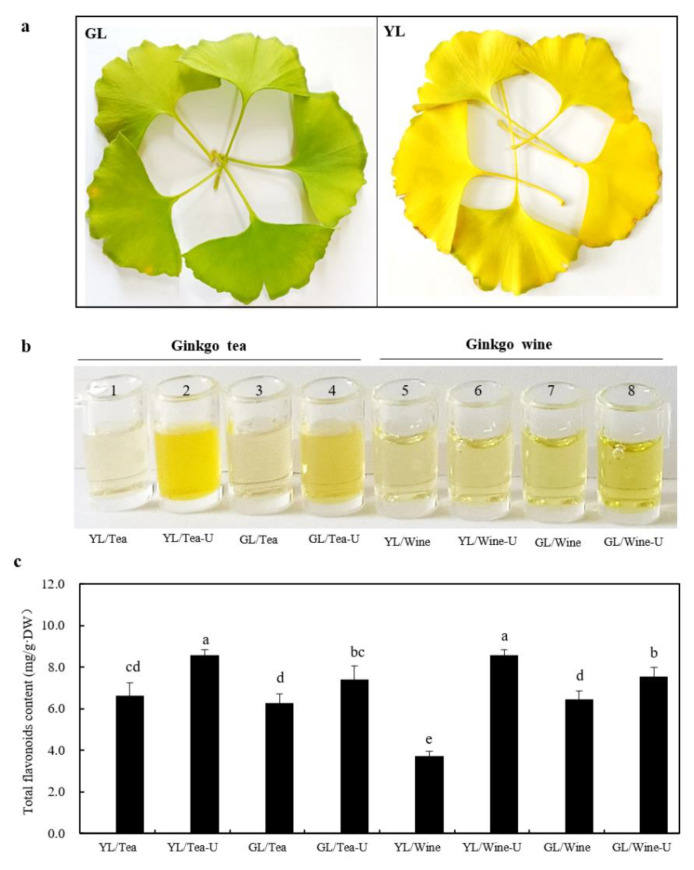
The raw materials, appearance and total flavonoid contents of the ginkgo tea and wine. (**a**) The appearance of the raw materials used for GBE drinks. Green ginkgo leaves (GL) were collected in late October (**Left**) and naturally newly senescent yellow leaves (YL) were collected in early November (**Right**). (**b**) The appearance of the prepared ginkgo tea (numbered 1–4) and ginkgo wine (numbered 5–8) that were derived from the GL and YL powder with or without ultrasound-assisted extraction. (**c**) Total flavonoid contents of ginkgo tea and ginkgo wine made from leaf powder. Rutin was used as the standard, and all the absorption values were obtained at 510 nm. Data are presented as mean ± SD. Different letters represent significantly different values with an alpha value at 0.05, *n* = 3.

**Figure 3 molecules-27-07479-f003:**
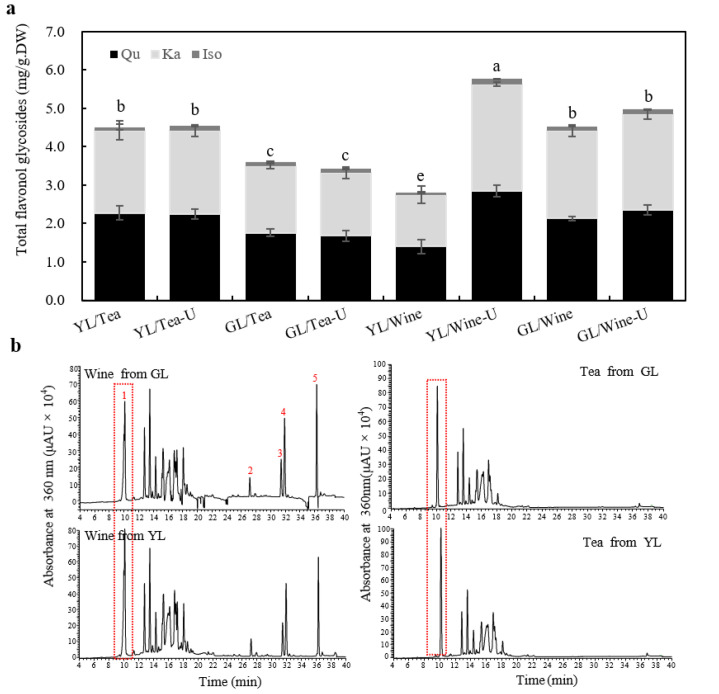
Analyses of the total flavonols in the ginkgo tea and ginkgo wine. (**a**) Total flavonol glycosides analyses of the ginkgo tea and ginkgo wine. Data are presented as mean ± SD. Different letters represent significantly different values with an alpha value at 0.05, *n* = 3. (**b**) HPLC chromatograms of ginkgo wine from GL powder (left, upper panel), YL powder (left, lower panel), ginkgo tea from GL powder (right, upper panel) and YL powder (right, lower panel) with ultrasound assistance. The red numbers in the chromatograms represent the peaks of specific compounds. (1) The specific age-related compound boxed at 9.94 min; (2) bilobetin; (3) ginkgetin, (4) isoginkgetin; (5) amentoflavone. Qu, quercetin-derived flavonol glycosides; Ka, kaempferol-derived flavonol glycosides; Iso, isorhamnetin-derived flavonol glycosides.

**Figure 4 molecules-27-07479-f004:**
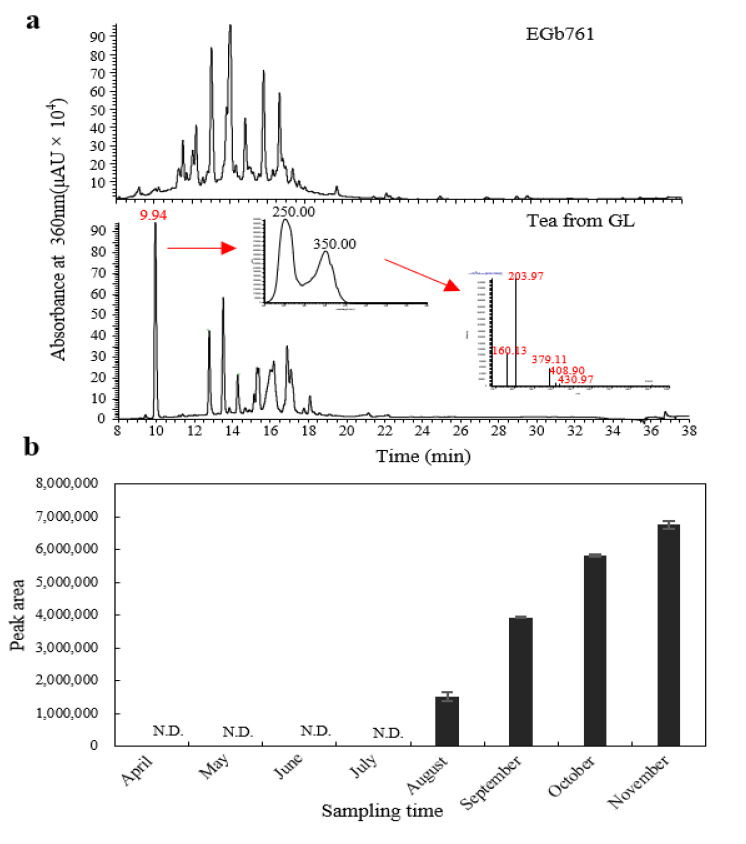
Characterization of the age-related compound during leaf development. (**a**) HPLC chromatograms of EGb761 (upper panel) and ginkgo tea from GL (lower panel) with the specific UV spectrum and mass spectra. (**b**) The accumulation profiles of the age-related compound at 9.94 min during leaf development from April to November. Data are presented as mean ± SD. *n* = 3.

**Figure 5 molecules-27-07479-f005:**
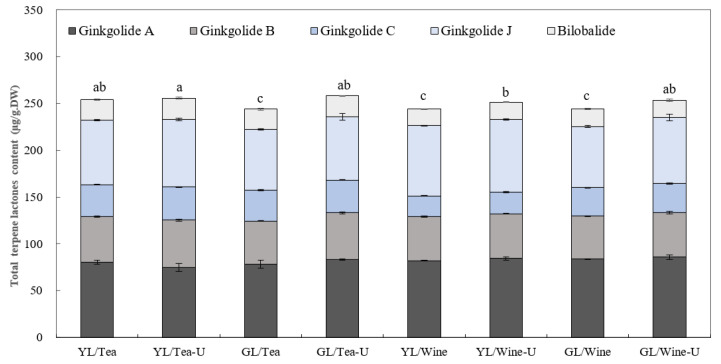
Analyses of the terpene trilactones (TTLs) in the ginkgo tea and wine. TTLs were quantified and calculated with the specific standard curve plotted by UHPLC-QTOF-MS with authentic substrates. Data are presented as mean ± SD. Different letters represent significantly different values with an alpha value at 0.05, *n* = 3.

**Figure 6 molecules-27-07479-f006:**
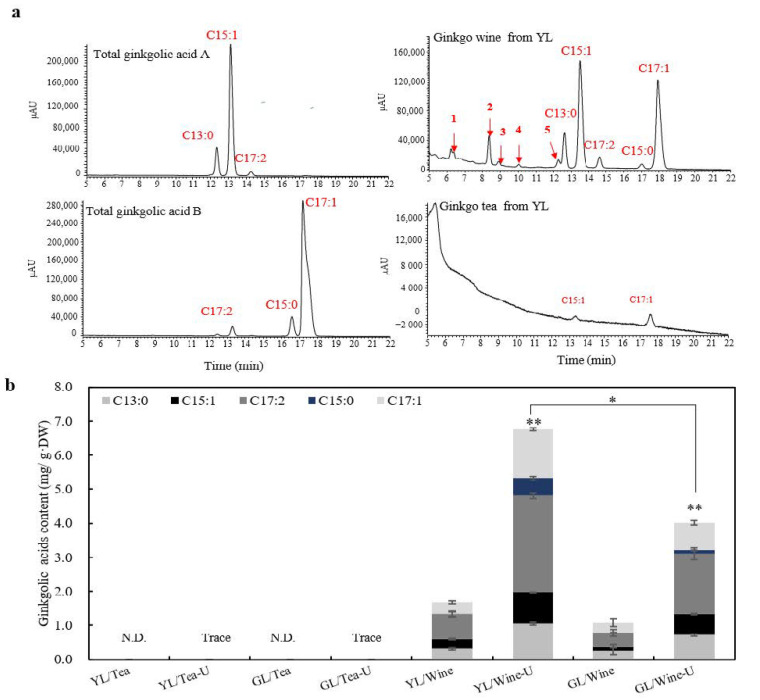
Characterization of the total ginkgolic acids in ginkgo tea and ginkgo wine. (**a**) HPLC chromatograms of ginkgo wine (right, upper panel) and ginkgo tea (right, lower panel) infusions for analyzing the total ginkgolic acids with authentic total ginkgolic acids A (left, upper panel) and B (left, lower panel). The red numbers 1–5 in the chromatograms represented the suspected newly identified ginkgolic acids peaks. (**b**) Quantification of the total ginkgolic acids in ginkgo tea and ginkgo wine. Ginkgolic acids were quantified and calculated with the specific standard curve plotted by HPLC with authentic substrates. Data are presented as mean ± SD. Asterisks denote *t*-test significance: * *p* < 0.05, ** *p* < 0.01, *n* = 3.

**Figure 7 molecules-27-07479-f007:**
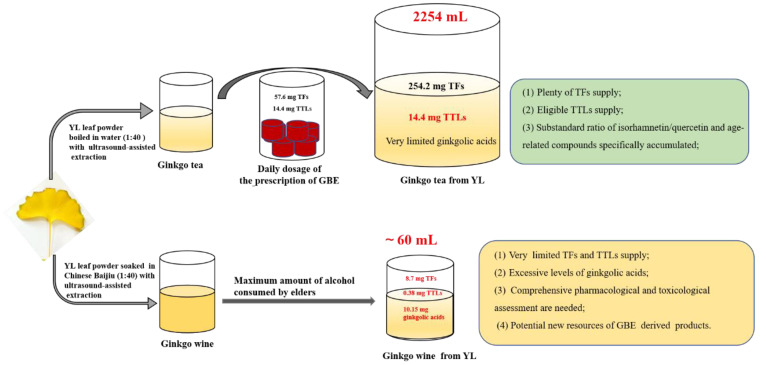
Summary of ingredient composition of the ginkgo tea and ginkgo wine compared with the ADI (acceptable daily intake) of prescription drugs “Yinxingye Pian” (with an SFDA approval number Z20053912). The ADIs of ginkgo flavonol glycosides (TFs) and terpene trilactones (TTLs) for disease treatment are stipulated as 57.6 mg and 14.4 mg according to the drug instructions, respectively. In our study, YL powder was infused with a solid-to-liquid ratio of 1:40. Then, 2254 mL of the first tea infusion contained 254.2 mg of TFs and 14.4 mg of TTLs. In addition, about 60 mL of the first ginkgo wine infusion contained 8.7 mg of TFs, 0.38 mg of TTLs, 10.15 mg of ginkgolic acids and plentiful biflavones.

**Table 1 molecules-27-07479-t001:** Antioxidant properties of GBE drinks.

	DPPH (TE)	ABTS (TE)
YL/Tea	129.27 (0.55) c	525.86 (11.55) cd
YL/Tea-U	133.59 (0.26) a	578.95 (4.49) abc
GL/Tea	125.48 (0.15) d	518.14 (7.06) cd
GL/Tea-U	126.71 (0.46) d	520.71 (12.09) d
Average/Tea	128.76 (3.26)	534.23 (26.69)
YL/Wine	132.89 (0.15) a	633.25 (15.78) a
YL/Wine-U	133.23 (0.31) a	612.38 (10.98) ab
GL/Wine	131.83 (0.15) ab	565.49 (9.31) bcd
GL/Wine-U	130.42 (0.26) bc	534.63 (3.67) cd
Average/Wine	132.09 (1.16)	586.43 (41.45)
Water	3.10 (1.31) f	171.31 (6.44) e
Chinese Baijiu	12.93 (0.61) e	194.30 (16.29) e

Note: The data are presented as the means of three replicates (S.D.). Different letters represent significantly different values. DPPH and ABTS are expressed as mg/L trolox equivalents (TE).

**Table 2 molecules-27-07479-t002:** Measurement of the total flavonol (TF) concentrations in GBE drinks.

	Quercetin (mg/g·DW)	Kaempferol (mg/g·DW)	Isorhamnetin (mg/g·DW)	Total Flavonols (mg/g·DW)
YL/Tea	2.27 (0.18) b	2.15 (0.24) c	0.09 (0.07) ab	4.51 (0.41) b
YL/Tea-U	2.25 (0.13) b	2.17 (0.15) c	0.13 (0.01) a	4.54 (0.29) b
GL/Tea	1.76 (0.09) c	1.74 (0.06) d	0.11 (0.01) ab	3.61 (0.16) c
GL/Tea-U	1.68 (0.14) c	1.63 (0.15) d	0.10 (0.01) ab	3.42 (0.29) c
Average/Tea	1.99 (0.31)	1.92 (0.29)	0.11 (0.03)	4.02 (0.59)
YL/Wine	1.40 (0.18) d	1.34 (0.23) e	0.06 (0.01) b	2.81 (0.43) e
YL/Wine-U	2.84 (0.15) a	2.78 (0.06) a	0.14 (0.00) a	5.76 (0.21) a
GL/Wine	2.13 (0.06) b	2.28 (0.15) bc	0.12 (0.01) ab	4.53 (0.21) b
GL/Wine-U	2.35 (0.13) b	2.50 (0.13) b	0.12 (0.00) a	4.97 (0.26) b
Average/Wine	2.27 (0.55)	2.31 (0.55)	0.11 (0.03)	4.69 (1.12)

Note: U represents ultrasound-assisted extraction. Data are represented as mean (S.D.). *n* = 3. Different letters represent significantly different values (*p* < 0.05).

**Table 3 molecules-27-07479-t003:** Measurement of the total terpene trilactones (TTLs) concentrations in GBE drinks.

	Ginkgolide A (μg/g·DW)	Ginkgolide B (μg/g·DW)	Ginkgolide C (μg/g·DW)	Ginkgolide J (μg/g·DW)	Bilobalide (μg/g·DW)	Total TTLs (μg/g·DW)
YL/Tea	80.59 (2.22) bc	48.62 (0.86) bc	34.01 (0.45) b	69.12 (0.65) cd	21.61 (0.77) a	253.96 (2.51) ab
YL/Tea-U	74.89 (4.34) d	50.60 (1.23) a	35.56 (0.03) a	71.81 (1.21) bc	22.71 (1.02) a	255.58 (4.74) a
GL/Tea	78.57 (4.12) cd	45.87 (0.46) e	32.93 (0.46) c	64.71 (0.65) e	21.81 (0.76) a	243.89 (3.99) c
GL/Tea-U	83.52 (0.84) ab	49.63 (0.98) ab	35.06 (0.40) a	67.41 (3.41) de	22.49 (0.43) a	258.13 (1.67) ab
Average/Tea	79.39 (4.28)	48.68 (2.01)	34.39 (1.11)	68.27 (3.13)	22.16 (0.82)	252.89 (6.36)
YL/Wine	82.07 (0.42) abc	46.95 (0.71) de	22.26 (0.32) f	75.03 (0.25) ab	17.71 (0.44) b	244.01 (1.15) c
YL/Wine-U	84.46 (1.62) ab	47.90 (0.27) cd	22.89 (0.59) f	77.00 (0.28) a	18.86 (0.46) b	251.52 (2.60) b
GL/Wine	83.30 (0.53) ab	45.85 (0.38) e	30.35 (0.34) e	65.49 (1.23) e	18.61 (0.50) b	244.09 (1.28) c
GL/Wine-U	85.84 (2.24) a	47.48 (1.34) cd	30.97 (0.59) e	70.55 (3.52) cd	18.62 (1.28) b	253.46 (3.99) ab
Average/Wine	84.04 (1.87)	47.04 (1.05)	26.62 (4.26)	72.12 (5.02)	18.45 (0.79)	248.27 (4.96)

Note: U represents ultrasound-assisted extraction. Data are represented as mean (S.D.). Different letters represent significantly different values (*p* < 0.05). *n* = 3.

**Table 4 molecules-27-07479-t004:** Summary of the five deduced newly identified ginkgolic acids in ginkgo wine.

Peak	Retention Time (min)	UV Absorption Spectrum (nm)	*m*/*z* (ESI^-^)	Deduced Chemical Formula	Deduced Alkyl Substituents
1	6.19	245, 309	361.21	C_23_H_38_O_3_	C16:0
2	8.16	246, 309	389.20	C_25_H_42_O_3_	C18:0
3	8.71	248, 313	387.14	C_25_H_40_O_3_	C18:1
4	9.84	251, 313	317.05	C_20_H_30_O_3_	C13:1
5	12.02	251, 311	369.12	C_24_H_34_O_3_	C17:3

Note: The deduced structures are shown in Appendix A.

## Data Availability

Data are contained within the article.

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
