# Peer review of "Medicinal Values and Potential Risks Evaluation of Ginkgo biloba Leaf Extract (GBE) Drinks Made from the Leaves in Autumn as Dietary Supplements"

_molecules, 2022, doi:10.3390/molecules27217479_

Round 1

Reviewer 1 Report

This manuscript describes the evaluation of medicinal values and risks of Ginkgo biloba leaf extract drinks made from the leaves in autumn. The research measured the total flavonols and total terpenes trilactones in GBE drinks. The total flavonoids were determined using chemical colorimetry methods. They also quantified the terpene trilactones by UHPLC-QTOF-MS modified from method reported by Liu and co-authors (ref. 32). Overall, this is an interesting paper and could attract the interest of readers of molecules. I recommend publishing after some minor revisions:

1.       Provide Chemdraw structures of flavonoids, terpene trilactones in the introduction section where these molecules were first mentioned in the manuscript. Most of the readers may not necessary be Organic Chemist. Providing these structures will help readers to follow the paper.

2.       The English grammar and writing need to be improved.

Author Response

Thanks a lot for your kind comments, and our replies are in the attachment.

Reviewer 2 Report

The manuscript mainly investigated the total flavonoids content, terpenes trilactones and ginkgolic acids in ginkgo tea and ginkgo wine made from Ginkgo biloba leaf extract. The antixoidantive activites of the extract were also evaluated with DPPH and ABTS free radical scanvenging methods. The experements were well performed and the results were also rationaly interpretted. The study gave us new information on the the healthcare effects of the ginkgo tea and ginkgo wine. In my opinion, the manuscript can be published after some major issues being addressed:

1\Figure 2 should be centered.

2\ Figure 3 should delete the border on the left.

3\Figure 4 should delete the border in the bottom.

4\ The antioxidantive capacity can not reflect the potential healthcare capacity, so as the total flavonoids content. The expressions in line 306-307 should be adjusted.

5\ From the Figure S2, I can not understand the content detection of the terpenes trilactones, since ginkgolide A and ginkgolide B had almost the same retention times.

Author Response

(The authors gave the same response as above.)

Reviewer 3 Report

The article was well written and good research.

 pharmacopeia should be written as pharmacopoeia.

Page number 5 line 171, aglycones were listed instaed of list.

The author should justify the standardization of dosage of the Ginko drinks also.

Author Response

(The authors gave the same response as above.)

Round 2

Reviewer 2 Report

accept